# RETHINKING OPTIMAL TRANSPORT IN OFFLINE REINFORCEMENT LEARNING

## ABSTRACT

We present a novel approach for offline reinforcement learning that bridges the gap between recent advances in neural optimal transport and reinforcement learning algorithms. Our key idea is to compute the optimal transport between states and actions with an action-value cost function and implicitly recover an optimal map that can serve as a policy. Building on this concept, we develop a new algorithm called Extremal Monge Reinforcement Learning that treats offline reinforcement learning as an extremal optimal transport problem. Unlike previous transport-based offline reinforcement learning algorithms, our method focuses on improving the policy beyond the behavior policy, rather than addressing the distribution shift problem. We evaluated the performance of our method on various continuous control problems and demonstrate improvements over existing algorithms.

## 1 INTRODUCTION

Recently, deep reinforcement learning (RL) has shown remarkable progress in complex tasks such as strategy games (Vinyals et al., 2019), robotics (Mnih et al., 2015), and dialogue systems (Ouyang et al., 2022). Standard RL algorithms interact with the environment and receive rewards that are used to update the policy. However, in some areas, such as healthcare, industrial control, and robotics, implementing RL algorithms that interact with the environment can be costly and risky. To avoid these risks, offline RL algorithms utilize historical data of expert interactions with the environment. In offline settings, only a dataset of experiences is provided, and the objective is to learn an effective policy without any online interactions (Levine et al., 2020).

Standard actor-critic methods (Sutton & Barto, 2018) have demonstrated poor performance in offline settings due to the instability of the critic function when dealing with out-of-distribution actions sampled by the learned actor (Fujimoto et al., 2018). To avoid this issue, a different behavior cloning objective is used, which keeps the actor close to the actions on which the critic was trained. However, the provided offline data is often sub-optimal, and offline reinforcement learning algorithms are used to find a policy more efficient than that provided in the data (Kumar et al., 2022a). In offline learning, the goal is to find a *trade-off* between staying close to the behavior policy for stable out-of-distribution evaluation and improving upon the behavior policy.

To enable RL algorithms to be trained on offline datasets, various methods incorporating Optimal Transport (OT) (Villani, 2008) into the training, have been proposed (Wu et al., 2019; Dadashi et al., 2020; Papagiannis & Li, 2020; Li et al., 2021; Haldar et al., 2022; Cohen et al., 2021; Luo et al., 2023). OT is particularly relevant for addressing distribution shift problems by minimizing deviation from the behavior policy (§2.4).

In our paper, we rethink OT as a framework for offline reinforcement learning that aims at policy improvement, rather than avoiding distribution shift. Unlike previous OT-based approaches in RL, we do not introduce a new OT-based regularization or reward function (Dadashi et al., 2020; Papagiannis & Li, 2020). Instead, we propose **a novel perspective that views the entire offline RL problem as an optimal transport problem** between state and action distributions (§3.1). By utilizing the action-value function as a transport cost function and the policy as an optimal transport map, we formalize offline reinforcement learning as a *maximin* OT optimization problem.

Our formulation naturally allows to apply rich OT theory to the reinforcement learning. As example, we integrate recently proposed Extremal Optimal Transport (Gazdieva et al., 2023) into offline

reinforcement learning (§3.2). In context of the unpaired image translation, extremal optimal transport offers the theoretically best possible translation with respect to a given cost function (Gazdieva et al., 2023). By employing an RL-based cost function, extremal transport implicitly trains the map (policy) to identify the best possible action for a given state with respect to the cost function, and to improve the policy beyond the behavior policy, see **proposition 3.1)**.

**Contribution:** We propose a novel OT-based formulation for offline reinforcement learning (§3.1). Building on this formulation, we introduce a new algorithm that employs extremal optimal transport techniques to derive the most effective policy from the dataset, which outperforms the behavior policy (§3.2). To evaluate our methodology, we conducted various experiments using the D4RL benchmark suite (Fu et al., 2020). Our results demonstrate that the proposed approach not only outperforms other OT-based offline RL methods but also consistently improves the performance of current state-of-the-art offline RL techniques (§ 4).

## 2 BACKGROUND AND RELATED WORK

### 2.1 REINFORCEMENT LEARNING

Reinforcement learning is a well-established framework for decision-making processes in environments modeled as Markov Decision Processes (MDPs). An MDP is defined by the state space $S$, action space $A$, and their respective distributions $\mathcal{S}$ and $\mathcal{A}$. It also includes the reward function $r(s, a)$, transition distribution between states $T(s' \mid s, a)$, and discount factor $\gamma$. The goal in RL is to find a policy $\pi(a|s)$ that maximizes the expected cumulative reward over time $t$: $\mathbb{E}_\pi \left[ \sum_{t=0}^{H} \gamma^t r(s_t, a_t) \right]$, where $H$ can be infinity. The action-value function $Q^\pi$ is a crucial component in the reinforcement learning framework, as it provides an estimate of the expected cumulative reward that can be obtained by following a given policy $\pi$. This can be written mathematically as:

$$Q(s, a) = r(s, a) + \gamma \mathbb{E}_{s' \sim T(s,a), a' \sim \pi(s')}[Q(s', a'))]. \tag{1}$$

where $s'$ and $a'$ are the next state and the next action respectively. This recursive formulation allows the action-value function $Q$ to be estimated by iteratively updating its estimate until convergence is reached. In continuous state spaces, finding the exact optimal policy is intractable due to the curse of dimensionalityy (Sutton & Barto, 2018). To address this, the popular Actor-Critic algorithm can be used (Sutton & Barto, 2018). Actor-Critic includes two components: an action-value function $Q(s, a)$ and a policy function $\pi$. In practice, $Q$ is updated by minimizing the mean squared Bellman error over the experience replay dataset $(s, a, s', r)$:

$$\min_Q \mathbb{E}_{(s,a,s') \sim \mathcal{D}} \left[ \left( r(s, a) + \gamma \mathbb{E}_{a' \sim \pi_\theta(s')} [Q(s', a')] - Q(s, a) \right)^2 \right]. \tag{2}$$

Accurate estimation of the action-value function allows a near-optimal policy to be recovered. For continuous control tasks, Deterministic Policy Gradient (DPG) (Silver et al., 2014) method can be employed. In this method, the actor is simply updated in the direction of the gradient of the action-value function with respect to the action:

$$\min_\pi \mathbb{E}_{s \sim \mathcal{D}} \left[ -Q(s, \pi_\theta(s)) \right]. \tag{3}$$

In combination with deep neural networks, the algorithm based on (3) is called Deep Deterministic Policy Gradient (DDPG) (Lillicrap et al., 2015). Its improved version, called Twin Delayed Deep Deterministic Policy Gradient (TD3) (Fujimoto et al., 2018), finds application in complex continuous control problems.

### 2.2 OFFLINE REINFORCEMENT LEARNING

Offline RL and behavior cloning (BC) approaches provide a fully dataset-driven RL that does not require interactions with the environment. The dataset $\mathcal{D}$ frequently contains expert as well as potentially irrelevant behaviors, and offline RL algorithms combine useful behavior segments spread across suboptimal trajectories (Levine et al., 2020). In contrast to standard RL, offline algorithms may face challenges due to *distribution shift* during the evaluation step. Specifically, distribution shift refers to the difference between the distribution of state-action pairs given by the expert $\beta$ and

the distribution of actions induced by the policy $\pi$ (Fujimoto et al., 2018). In general, there are two classes of algorithms designed to address these issues:

**Policy penalty:** To estimate the maximum $Q$-value over actions while constraining the learned policy to stay within the behavior data distribution's and avoid distribution shift, many methods have been proposed (Brandfonbrener et al., 2021; Kumar et al., 2019; Zhou et al., 2021; Chen et al., 2022; Kumar et al., 2022b; Fujimoto & Gu, 2021; Kostrikov et al., 2021). For instance, Fujimoto & Gu (2021) proposed a minimalist approach, TD3+BC, that combines TD3 (Fujimoto et al., 2018) with minimizing the supervised loss between policy and expert actions, and showed strong performance.

Another policy penalty method is one-step RL (Brandfonbrener et al., 2021). This approach involves maximizing the policy log probability for the actions provided in the dataset without off-policy evaluation. The method approximates the behavior $Q^\beta$-function and extracts the corresponding policy $\pi$ by weighting actions using the advantage function $A^\beta(s, a)$ (Sutton & Barto, 2018).

Implicit $Q$-Learning (IQL) (Kostrikov et al., 2021) is another method that does not employ off-policy evaluation. IQL approximates the policy improvement step by treating the state value function $V^\beta(s)$ as a random variable and taking a state-conditional upper expectile to estimate the value of the best actions in the state.

**Critic penalty:** An alternative approach to address challenges in offline RL is to penalize the $Q$ function. The conventional $Q$ learning technique for offline RL uses (2), but with data $\mathcal{D}$ collected according to an expert policy $\beta$. To avoid overestimation bias (Fujimoto et al., 2018) during critic learning in offline settings, Policy Gradient from Arbitrary Experience via DICE (Nachum et al., 2019) proposes to incorporate $f$-divergence regularization into the critic function.

Based on a similar concept, Conservative $Q$-Learning (CQL) (Kumar et al., 2020) and ATAC Cheng et al. (2022) apply an adversarial training of the citric by extending the loss with additional terms that minimize the $Q$ values on the samples of a learned policy and maximize the values of the dataset actions. Another notable offline reinforcement learning method is Fisher Behavior Regularized Critic (F-BRC) (Kostrikov et al., 2021). The F-BRC technique augments the standard Bellman critic loss with a regularization based on Fisher divergence.

## 2.3 OPTIMAL TRANSPORT

Monge's problem (Villani, 2008) was the first example of the OT problem. Monge's formulation of OT with a cost function $c$ aims at finding a mapping $T$ of the two probability distributions $\mu$ and $\nu$, where $T\sharp\mu = \nu$ is the mass-preserving condition and $T\sharp$ is the push-forward operator [§1](Villani, 2008):

$$\min_{T\sharp\mu=\nu} \mathbb{E}_{x\sim\mu}[c(x, T(x))]. \tag{4}$$

Informally, we can say that the cost is a measure of how hard it is to move a mass piece between points $x \in \mathcal{X}$ and $y \in \mathcal{Y}$ from distributions $\mu$ and $\nu$ correspondingly. That is, an OT map $T$ shows how to optimally move the mass of $\mu$ to $\nu$, i.e., with the minimal effort.

A widely recognized alternative formulation for optimal transport was introduced by Kantorovich (Kantorovitch, 1958). Unlike the Monge's OT problem formulation, this alternative allows for mass splitting. It is important because for certain values of $\mu$ and $\nu$, there may not be a map $T$ satisfying $T_\#\mu = \nu$ in the Monge's OT (4). The Kantorovich OT problem can be written as:

$$\min_{\gamma\in\Pi(\mu,\nu)} \mathbb{E}_{(x,y)\sim\gamma}[c(x, y)]. \tag{5}$$

In this case, the minimum is obtained over the transport plans $\gamma$, which refers to the couplings $\Pi$ with the respective marginals being $\mu$ and $\nu$. The optimal $\gamma^*$ belonging to $\Pi(\mu, \nu)$ is referred to as the *optimal transport plan*. The dual form of (5) following (Villani, 2003) can be written as:

$$\max_f \left[ \mathbb{E}_{x\sim\mu}[f^c(x)] + \mathbb{E}_{y\sim\nu}[f(y)] \right], \tag{6}$$

where $\max$ is taken over all $f \in \mathcal{L}^1(\nu)$, and $f^c(x) = \min_y [c(x, y) - f(y)]$ is $c$-transform of $f$ (Villani, 2008). For cost functions $c(x, y) = \|x - y\|_2$, the solution of the optimization problems (4, 5) is called the *Wasserstein-1* distance, see (Villani, 2008, §1) or (Santambrogio, 2015, §1, 2). In recent years, significant progress has been achieved in utilizing neural networks for computing OT maps to

address strong (Makkuva et al., 2019; Korotin et al., 2021; Gulrajani et al., 2017), weak (Korotin et al., 2022c), and general (Asadulaev et al., 2022) OT problems. Our work is inspired by these developments, as we leverage neural optimal transport methods to improve offline RL.

## 2.4 OPTIMAL TRANSPORT IN OFFLINE REINFORCEMENT LEARNING

The simplest OT-based approach to avoid distribution shift is the behavior regularized actor-critic (BRAC) (Wu et al., 2019). This method is based on a behavior-regularized actor-critic that includes a different divergence $D$ and design choices for offline RL approaches. In general, the BRAC framework can be written as *distance regularized* policy optimization:

$$\min_{\pi} \mathbb{E}_{s\sim\mathcal{D},a\sim\pi(s)} \left[-Q\left(s,a\right)\right] + \alpha \mathbb{E}_{s\sim\mathcal{D}}[D(\pi(\cdot|s),\beta(\cdot|s))]. \tag{7}$$

If we put the *Wasserstein-1* as a divergence $D$, we obtain W-BRAC algorithm. The core idea is to use the OT distance to measure the difference between the policy $\pi$ and the expert policy $\beta$. Given the action-value function $Q(s,a)$, the discriminator $f$, and the behavior cloning impact coefficient $\alpha$, we have in (7):

$$\min_{\pi} \max_{\|f\|_L \leq 1} \left[ \underbrace{\mathbb{E}_{s\sim\mathcal{D},a\sim\pi(s)}[-Q(s,a)]}_{\text{Critic function}} + \alpha \left( \underbrace{\mathbb{E}_{s\sim\mathcal{D},a\sim\beta(\cdot|s)}[f(a)] - \mathbb{E}_{s\sim\mathcal{D},a\sim\pi(\cdot|s)}[f(a)]}_{\textit{Wasserstein-1}\text{ distance}} \right) \right] \tag{8}$$

The problem is that finding the analytical solution to the dual *Wasserstein-1* formulation typically *is intractable*, and the potential $f$ is required to satisfy the *1-Lipschitz* condition. It was shown that this optimal transport method has drawbacks in computing *Wasserstein-1* distance in practice (Korotin et al., 2022a). Also, it is important to note that the coefficient $\alpha$ should be carefully chosen.

Another widely explored way to apply optimal transport in offline RL is to use it to construct a *pseudo-reward* function (Dadashi et al., 2020; Papagiannis & Li, 2020; Li et al., 2021; Haldar et al., 2022; Cohen et al., 2021). Primal Wasserstein Imitation Learning (Dadashi et al., 2020) and Sinkhorn Imitation Learning (Papagiannis & Li, 2020) utilizing OT distance between the expert and imitator to create a reward function. Recently, the Optimal Transport Reward Labeling (OTR) method (Luo et al., 2023) was proposed to generate a reward-annotated dataset, which can then be employed by various offline RL algorithms. However, as the W-BRAC these methods use OT primarily as an additional *loss* function. In summary, we can say that **the main aspects** of the existing OT in RL methods are:

- 1) Wasserstein regularized actor-critic requires *1-Lipschitz* function constraints on $f$, which complicates the optimization.
- 2) Optimal transport incorporated in RL with a coefficient $\alpha$, and it was shown that the choice of $\alpha$ value is crucial (Brandfonbrener et al., 2021).
- 3) Previously proposed OT approaches focuses on behavior cloning and does not address policy improvement beyond the behavior policy.

In our paper, we tackle the challenge of developing an OT-based approach that avoids these limitations. In the following section, we present our *maximin* OT as RL formulation and formally compare it to the W-BRAC framework.

## 3 METHOD

### 3.1 MONGE REINFORCEMENT LEARNING

To rethink application of OT in RL, lets consider the (4). If we replace cost $c(x,y)$ with the optimal $Q(s,a)$ function for the given data, distribution $\mu$ with the state distribution given by the expert $\mathcal{S}$, and distribution $\nu$ with the expert actions distribution $\mathcal{A}$, we obtain a following problem:

$$\min_{\pi\sharp\mathcal{S}=\mathcal{A}} \mathbb{E}_{s\sim\mathcal{S}}[-Q(s,\pi(s))]. \tag{9}$$

If policy $\pi$ is deterministic, we can view this formulation as the DPG algorithm (3) with *distributional constraints* on the policy $\pi$. To convert the problem into a one that can be applied to deep RL and solved via neural network approximations (Fan et al., 2021), we can consider a dual form of (5).

To do this, we expand the c-transform of the dual form in (6) using the function $Q$ as the cost $c$, which can be represented as $\mathbb{E}_{s\sim\mathcal{S}}[f^Q(s)] = \mathbb{E}_{s\sim\mathcal{S}}[\min_a \{-Q(s,a) - f(a)\}]$. Subsequently, by applying the Rockafellar interchange theorem (Rockafellar, 1976, Theorem 3A), we replace the optimization over points $a \in \mathcal{A}$ with an equivalent optimization over functions $\pi : \mathcal{S} \to \mathcal{A}$:

$$\mathbb{E}_{s\sim\mathcal{S}}[f^Q(s)] = \min_\pi \mathbb{E}_{s\sim\mathcal{S},a\sim\pi(s)}[-Q(s,a) - f(a)]. \tag{10}$$

After substituting (10) with (6), we get:

$$\max_f \min_\pi \left[ \underbrace{\mathbb{E}_{s\sim\mathcal{S},a\sim\pi(s)}[-Q(s,a)]}_{\text{Cost function}} - \underbrace{\mathbb{E}_{a\sim\pi(s)}[f(a)] + \mathbb{E}_{a\sim\mathcal{A}}[f(a)]}_{\text{Constraints preserving}} \right] \tag{11}$$

Let's denote the expression in the large square brackets by $\mathcal{L}(f,\pi)$. Then, following (Korotin et al., 2022c) for every optimal potential $f^* \in \arg\max_f \left[ \min_{\pi:\mathcal{S}\to\mathcal{A}} \mathcal{L}(f,\pi) \right]$, we can solve a saddle point problem in (11) and extract an OT map $\pi^*$ between $\mathcal{S}, \mathcal{A}$ from the optimal pair $(f^*, \pi^*)$.

We call this method *Monge Reinforcement Learning* (MRL). This formulation provides a *maximin* optimal transport problem, where the optimal policy is defined as an optimal mapping from states to actions with an action-value cost function. This optimization problem instantly bypass the *1-Lipschitz* function constraints on $f$. The specific role of the potential $f$ in the objective of (11) is different from its function in (8). Here, the $f$ is used to verify that the constraints on the optimization have been satisfied, rather than to compute the *Wasserstein-1* distance itself.

More formally, the potential $f$ can be seen as a *Lagrangian multiplier* (Peyre, 2021) that ensures that the generated distribution $\pi\sharp S$ matches the distribution of actions given by the dataset $\mathcal{D}$, and penalizes the former if it does not. This formulation provides a foundation for OT in RL and compared to the previously proposed W-BRAC method, our method has the following advantages

- 1) It does not require *1-Lipschitz* function constraints on $f$. Instead, the function $f$ plays the role of a *Lagrangian multiplier*.
- 2) There is no dependence on the hyperparameter $\alpha$, which controls the impact of optimal transport *loss* during policy optimization. The whole OT problem is aimed at finding an optimal policy.

However, in (11) the policy maps the distribution of states to the *complete* distribution of actions. In offline reinfrocement learning, the provided dataset often contains suboptimal paths, and the policy should apply *outlier detection* in the action space to avoid inefficient actions for each state in the dataset. In the following subsection, we introduce an algorithm that avoids inefficient actions using optimal transport.

## 3.2 EXTREMAL MONGE REINFORCEMENT LEARNING

An efficient offline RL algorithm should be able to *stitch* between trajectories for each state provided by the dataset (Kumar et al., 2022a). To do this, the policy should implement outlier detection in the action space and select only the effective actions from the mixture of expert demonstrations. More formally, we can say that the goal is to identify a policy that maps into the support of the action space, $\pi(s) \in \text{supp}(\mathcal{A})$, that maximizes $Q(s,a)$ for each state. Here support is the set of points where the probability mass of the distribution lives.

Recently, the OT formulation with similar constraints on the transport map was considered in the Extremal Optimal Transport (ET) method (Gazdieva et al., 2023). The ET formulation represents the theoretically best possible unpaired domain translation based on a given cost function. Essentially, for the Euclidean cost functions as $\ell^2$, the extremal transport maps can be seen as the tool for finding *nearest neighbors* (maximally close) to the input samples from the target, according to the cost function, see Figure 1 in (Gazdieva et al., 2023).

In the context of RL, we can say that the extremal map will align the state space with the actions that maximize the $Q$ cost function. Formally, for RL, we can express the optimization problem as:

$$\min_{\pi\sharp\mathcal{S}\subset\text{Supp}(\mathcal{A})} \mathbb{E}_{s\sim\mathcal{S}}[-Q(s,\pi(s))]. \tag{12}$$

In this case, the $\min$ is taken over a function $\pi : \mathcal{S} \to \mathcal{A}$ that maps from $\mathcal{S}$ to the support of $\mathcal{A}$, denoted as $\mathrm{Supp}(\mathcal{A})$. The goal is to find such a support, $\mathrm{Supp}(\mathcal{A})$, that minimizes the expectation $\mathbb{E}_{s \sim \mathcal{S}}[-Q(s, \pi(s))]$. The main difference between this and the optimal transport problem discussed earlier in (9), lies in the constraints used. The previous method used *equality* constraints, while this technique imposes *inclusion* constraints, allowing only *partial* alignment between state and action spaces. See, visual representation of the Extremal Monge RL in Figure 1.

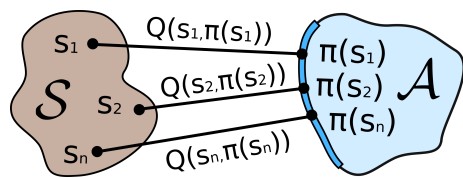

Figure 1: *Extremal Monge RL problem*: Learned transport map $\pi$ aligns states $\mathcal{S}$ to a subset of the experts actions $\mathcal{A}$, which maximize the critic-based cost function $Q(s, \pi(s))$.

To show that (12) allows to improve the policy $\pi$ beyond the behavior policy $\beta$, let's consider a one-step RL algorithm. Starting from the expert policy $\beta$, we solve the extremal transport problem (12) with respect to the given cost function $Q^{\beta}$.

**Proposition 3.1** (Policy Improvement with Extremal Transport). *For any policy $\pi$, let's define its performance as $J(\pi) \overset{def}{=} \underset{\tau \sim \pi}{\mathrm{E}}[\sum_{t=0}^{\infty} \gamma^t r(s_t, a_t, s_{t+1})]$ where $\tau \sim \pi$ indicates the random trajectories obtained by following the policy $\pi$, $(s_0 \sim S, a_t \sim \pi(s_t), s_{t+1} \sim P(\cdot \mid s_t, a_t))$. Let $\beta$ be the policy of an expert and $\pi$ is the solution to (12) with the $Q^{\beta}$ cost function. Then it holds that: $J(\pi) \geq J(\beta)$.*

The proof is given in Appendix (§A). With this proposition, we show that we avoid a problem 3 mentioned in (§2.4) by proposing an OT for policy improvement beyond behavioral policy.

To obtain a saddle-point reformulation that can be used to compute the solution to (12) in practice, the steps analogous to those presented in (§3.1), can be considered (Gazdieva et al., 2023). This reformulation replaces set *inclusion* constraints with *inequality* constraints, and if the parameter $w \to +\infty$, it provides an extremal transport. This leads to the following optimization problem:

$$\max_{f \geq 0} \min_{\pi} \mathbb{E}_{s \sim \mathcal{S}, a \sim \pi(s)}[-Q(s, a)] - \mathbb{E}_{s \sim \mathcal{S}, a \sim \pi(s)}[f(a)] + w \mathbb{E}_{a \sim \mathcal{A}}[f(a)]. \quad (13)$$

The changes compared to (11) are emphasized by the green color. In the (13), the $w$ coefficient is used to control the extreme of the chosen action, and the potential $f$ is always greater or equal to 0. Here $f$ is checking that the constraints in the (12) are met. According to the (Gazdieva et al., 2023, Proposition 4) the optimal potential $f^*$ vanishes on the outliers, i.e. $f^*(a) = 0$ holds for all $a \in \mathrm{Supp}(\mathcal{A}) \setminus \mathrm{Supp}(\pi^* \sharp \mathcal{S})$. It is important that the algorithm defines outliers by the critic function $Q^{\beta}(s, a)$, because the optimization goal is to find a $\mathrm{Supp}(\mathcal{A})$ for which the value $-Q$ is minimum.

The policy update in (13), in short can be written as $\min_{\pi} \mathbb{E}_{s \sim \mathcal{D}, a \sim \pi(s)}[-Q(s, a) - f(a)]$. Here we can see that if $f(a) = 0$ only on the out-of-support actions, then it adds *extra weight* to the correct actions, indicating policy to choose actions in $\mathcal{A}$ that minimize the $-Q$ function. As a result, a map (policy) learned with extremal transport will ignore inefficient actions provided in dataset.

We can interpret that *our method lies between behavior cloning and direct maximization of the $Q$ function.* Recent studies have shown that direct maximization can lead to suboptimal results due to overestimation bias (Fujimoto et al., 2019). Conversely, being too close to the expert's policy prevents improvement (Kumar et al., 2022a). Intuitively, our extremal formulation allows us to strike a balance that maximizes $Q$ by taking actions from the expert action distribution that are not radically different from those on which the $Q$ function was trained. In the following section, we propose the optimization procedure for the Extremal Monge-RL (XMRL) algorithm .

### 3.3 PRACTICAL OPTIMIZATION PROCEDURE

In this section, we provide a practical solution for learning policies from offline data. While the optimal $Q(s, a)$ for the given data is unknown, we can approximate it by a neural network $Q_{\phi} : \mathbb{R}^S \times \mathbb{R}^A \to \mathbb{R}$, trained on the given expert data $\mathcal{D}$ via (2). If the $Q$ function is trained only on actions sampled from the behavior policy $\beta$, we get $Q^{\beta}$ and $Q^{\pi}$ when the next actions $a'$ are sampled from the learned policy $\pi$.

Unfortunately, in practice, the $Q$ function learned by (2) suffers from overestimation bias (Fujimoto et al., 2018; 2019; Fujimoto & Gu, 2021; Kumar et al., 2020; 2019; 2022a; Levine et al., 2020). In OT,

---

**Algorithm 1** Extremal Monge Reinforcement Learning (XMLR)

---

**Input:** Dataset $\mathcal{D}(s, a, r, s')$
Initialize $Q_\phi, \pi_\theta, f_\omega, \beta$
**for** $k$ in 1...N **do**
    $(s, a, r, s') \leftarrow \mathcal{D}$: sample a batch of transitions from the dataset.
    $Q^{k+1} \leftarrow$ Update cost function $Q_\phi$ using the Bellman update in (2) or (14)
    $\pi^{k+1} \leftarrow$ Update policy $\pi_\theta$ as a transport map using the objective in (15)
    $f^{k+1} \leftarrow$ Update $f_\omega$ using outputs of $\pi_\theta$ and samples from dataset, using (16)
**end for**

---

the cost function that does not match the given problem can lead to suboptimal results (Liu et al., 2020; Asadulaev et al., 2022). To avoid this issue, we tested our method in pair with the two well-known approach to obtain not overestimated critic: CQL (Kumar et al., 2020) and TD3+BC (Fujimoto & Gu, 2021). For the $k$ update steps, the conservative $Q$ function can be obtained by:

$$Q^{k+1} \leftarrow \arg \min_Q \mathbb{E}_{s \sim \mathcal{D}, a \sim \pi(s)} \left[ Q^k(s, a) \right] - \mathbb{E}_{s \sim \mathcal{D}, a \sim \beta(s)} \left[ Q^k(s, a) \right]$$
$$+ \frac{1}{2} \mathbb{E}_{(s, a, s') \sim \mathcal{D}} \left[ \left( r(s, a) + \gamma \mathbb{E}_{a' \sim \beta(s')} \left[ Q^k(s', a') \right] - Q^k(s, a) \right)^2 \right] + \mathcal{R}(\pi). \tag{14}$$

Where $\mathcal{R}(\pi)$ is a regularization, usually entropy based. For the TD3+BC based cost, we used the $Q$ function learned by (2), but coupled with the BC loss, this cost can be written as $c(s, \pi(s)) = -\lambda Q(s, \pi(s)) + (\pi(s) - a)^2$. The algorithm with the CQL-based cost (critic) we call **XMRL**$^{\mathbf{CQL}}$ and the algorithm with a BC-based cost we call **XMRL**$^{\mathbf{BC}}$. In general, *any RL-based cost* function can be used. The core of our method is the policy penalty, which can help to avoid suboptimal actions provided by the expert and not the critic learning.

To obtain a mapping $\pi$ from the distribution of states $\mathcal{S}$ to the distribution of actions $\mathcal{A}$, we use neural networks $\pi_\theta : \mathbb{R}^S \to \mathbb{R}^A$ and $f_\omega : \mathbb{R}^D \to \mathbb{R}$ to parameterize $\pi$ and $f$ respectively. These neural networks serve as function approximators that can capture complex mappings of states. For the $k$ update step, given the last policy $\pi^k$, the value function $Q^k$, and $f^k$, we have:

$$\pi^{k+1} \leftarrow \arg \min_\pi \mathbb{E}_{s \sim \mathcal{D}, a \sim \pi^k(s)} [-Q^k(s, a) - f^k(a)]. \tag{15}$$

$$f^{k+1} \leftarrow \arg \min_f -\mathbb{E}_{s \sim \mathcal{D}, a \sim \pi^k(s)} [f^k(a)] + w \mathbb{E}_{s, a \sim \mathcal{D}} [f^k(a)]. \tag{16}$$

By assigning a weight $w$, we can impose a more significant increase in the values of the highly rewarded actions. We train these neural networks using stochastic gradient ascent-descent (SGAD) by sampling random batches from the dataset $\mathcal{D}$. At each training step, we sample a batch of transitions from the offline dataset $\mathcal{D}$, and then adjust the value function $Q$, the policy $\pi$, and the potential $f$, see Algorithm 1. Our algorithm provides a scalable approach to learning policies from offline data, see the next section for practical evaluation.

## 4 EXPERIMENTS

### 4.1 BENCHMARKS AND BASELINES

We evaluate our proposed method using the Datasets for Deep Data-Driven Reinforcement Learning (D4RL) (Fu et al., 2020) benchmark suite, which is a collection of diverse datasets designed for training and evaluating deep RL agents in a variety of settings. It includes tasks in continuous control, discrete control, and multi-agent settings with a variety of reward structures and observation spaces. First, we tested our method on the Gym's `MuJoCo` environments, such as Walker, Hopper, and HalfCheetah (HalfC). Importantly, we also tested our method on the complex `Antmaze` environments.

To compare the performance of our proposed method, we selected four state-of-the-art offline RL algorithms. These include Conservative $Q$-Learning (CQL) (Kumar et al., 2020), Twin Delayed

Deep Deterministic Policy Gradient with behavior Cloning (TD3+BC) (Fujimoto & Gu, 2021), Implicit $Q$-Learning IQL (Kostrikov et al., 2021), and IQL with Optimal Transport Reward Labeling (IQL+OTR) (Luo et al., 2023). The results of the supervised behavior cloning are also included.

## 4.2 SETTINGS

**MuJoCo.** To be consistent with the theory given in (§3.2) For this environment, we tested how our proposed algorithm improves beyond the expert policy $\beta$. For this, following the setting provided by One-Step RL (Brandfonbrener et al., 2021), we pretrained $\beta$ for 500k steps using the provided data. Then we pre-trained $Q^\beta$ for 2 million steps. A two-layer feed-forward network with a hidden layer size of 1024 and a learning rate of 1e-4 was used with the Adam optimizer (Kingma & Ba, 2014). Then, we applied a *simplified* conservative update to $Q^\beta$ (see (Kumar et al., 2020)[Eq.1]). Using the next actions by $\beta$, this update provides a lower bound on the real $Q^\beta$ (Kumar et al., 2020)[Theorem.1]. The simplified version does not include the penalty term $\mathbb{E}_{s\sim\mathcal{D},a\sim\beta(s)}Q(s,a)$ and the regularization $\mathcal{R}(\pi)$. Then to improve beyond behavior policy $\beta$ we trained policy for the 100k steps using Algorithm 1 with a learning rate of 1e-5. The results are shown as XMRL$^{\text{CQL}}$ in Table 1.

For XMRL$^{\text{BC}}$, no pre-trained models were used, we trained TD3+BC from scratch for 1m steps in conjunction with our method. All hyperparameters were the same as the original (Fujimoto & Gu, 2021). See results in Table 1. For all experiments we set $w$ equal to 8. The ablation study on the parameter $w$ is given in the Appendix (§A.2).

**Antmaze.** While this environments are more complicated than MuJoCo, for the XMRL$^{\text{CQL}}$ we used *full* CQL update (14). No pre-trained models were used; we used the code provided by the CORL library (Tarasov et al., 2022), with all hyperparameters and architectures set identical to those originally proposed by CORL for the CQL method. We trained algorithm for 1m steps, with set $w$ equal to 8 for all experiments. See the results in Table 2.

For XMRL$^{\text{BC}}$, as a backbone we used the improved version of TD3+BC called ReBRAC (Tarasov et al., 2023), keeping the same hyperparameters proposed by the authors. The ReBRAC framework recommends task-specific hyperparameters, to achieve high results, thus we tested different values of $w$ for this algorithm. For the large-play, medium-diverse and umaze, it was equal to 5. For the medium-play, large-diverse, it was equal to 1, and for umaze-diverse, $w$ was set equal to 8. The results are shown in Table 2.

## 4.3 RESULTS

We present the qualitative results of our XMRL algorithms in Tables 1 2. For MuJoCo, the blue color in the tables highlights the top 3 scores, and **bold** indicates the best. We compared the results with those reported in the OTR+IQL paper (Luo et al., 2023). For the Antmaze, the main comparison is with the *oracle* to which our method is coupled, **bold** and blue indicates if the our algorithm is better than the *oracle*. In these experiments we reproduced CQL and ReBRAC results the code provided in CORL. The CQL score curves are provided in Appendix 4. For completeness, IQL and OTR+IQL results are also included in Table 2.

Our method shows promise for improving the field of offline RL using optimal transport. Importantly we show improvements over the previously best OT-based offline RL algorithm OTR+IQL. By introducing a novel mathematical task formulation, we provide effective solutions to complex RL problems. In offline RL, the problems are noisy, complex, and diverse, and task-specific hyperparameter search is required. But, we used the same set of parameters for each MuJoCo task, and our algorithm XMRL$^{\text{CQL}}$ is consistently in the top performing algorithms.

For the Antmaze, our method consistently outperforms the CQL on which it is based. For the XMRL$^{\text{BC}}$, our method provides state-of-the-art results for 4/6 environments.

Our formulation aligns the state space with only a *portion* of the action space. This property is particularly relevant for offline datasets that contain a *mixture* of expert demonstrations. Our method can help to select the best possible action from a set of expert trajectories (Levine et al., 2020). In other words, optimal $f$ is to encourage the discovery of new *extreme* actions that are atypical for the behavior policy at each state, but can yield high rewards.

Table 1: Averaged normalized scores on `MuJoCo` tasks. Reported scores are the results of the final 10 evaluations and 5 random seeds.

| | Dataset | BC | One-RL | CQL | IQL | OTR+IQL | TD3+BC | XMRL$^{CQL}$ | XMRL$^{BC}$ |
|---|---|---|---|---|---|---|---|---|---|
| Medium | HalfCheetah | 42.6 | 48.4 | 44.0 | 47.4 | 43.3 | 48.3 | **51.4±0.2** | 48.5±0.3 |
| | Hopper | 52.9 | 59.6 | 58.5 | 66.3 | 78.7 | 59.3 | **80.4±7.4** | 61.7±6.9 |
| | Walker | 75.3 | 81.1 | 72.5 | 78.3 | 79.4 | 65.5 | **84.3±0.6** | 83.1±3.6 |
| Medium-Replay | HalfCheetah | 36.6 | 38.1 | **45.5** | 44.2 | 41.3 | 44.6 | 44.8±0.5 | 44.9±0.6 |
| | Hopper | 18.1 | **97.5** | 95.0 | 94.7 | 84.8 | 60.9 | 92.1±10.8 | 85.5±6 |
| | Walker | 26.0 | 49.5 | 77.2 | 73.9 | 66.0 | 81.8 | **86.6±4.8** | 86.1±2.0 |
| Medium-Expert | HalfCheetah | 55.2 | 93.4 | 91.6 | 86.7 | 89.6 | 90.7 | 74.4±17.0 | **95.8±0.1** |
| | Hopper | 52.5 | 103.3 | 105.4 | 91.5 | 93.2 | 98.0 | **110.2±1.9** | 109.8±1.2 |
| | Walker | 107.5 | **113.0** | 108.8 | 109.6 | 109.3 | 110.1 | 111.2±0.7 | 110.7±1.2 |

Table 2: Averaged normalized scores on `Antmaze` tasks. Our algorithm XMRL outperforms prior methods on challenging tasks, which require choosing the best possible action provided by the different or sub-optimal policies. Reported scores are the results of the final 100 evaluations and 5 random seeds..

| Baseline | IQL | OTR+IQL | CQL (Oracle) | XMRL$^{CQL}$ | ReBRAC (Oracle) | XMRL$^{BC}$ |
|---|---|---|---|---|---|---|
| antmaze-umaze-v2 | 87.5 ± 2.6 | 83.4 ± 3.3 | 86.3 ±3.7 | **90±2.6** | 97.8 ± 1.0 | **98.0 ±1.7** |
| antmaze-umaze-diverse-v2 | 62.2 ± 13.8 | 68.9 ± 13.6 | 34.6 ±20.9 | **40±2.6** | 88.3 ± 13.0 | **92.0±2.3** |
| antmaze-medium-play-v2 | 71.2 ± 7.3 | 70.5 ± 6.6 | 63.0 ±9.8 | **67.3±10.1** | **84.0 ± 4.2** | 78.5 ±6.4 |
| antmaze-medium-diverse-v2 | 70.0 ± 10.9 | 70.4 ± 4.8 | 59.6 ±3.5 | **65.3±8.0** | 76.3 ± 13.5 | **90.6 ±2.0** |
| antmaze-large-play-v2 | 39.6 ± 5.8 | 45.3 ± 6.9 | 20.0 ±10.8 | **25.6±3.7** | **60.4 ± 26.1** | 52.3 ±31.8 |
| antmaze-large-diverse-v2 | 47.5 ± 9.5 | 45.5 ± 6.2 | 20.0 ±5.1 | **23.6 ±11.0** | 54.4 ± 25.1 | **59.6 ±6.0** |

However, our algorithm has a *limitation* in the setting of the hyperparameters $w$, as it was said, the value $w$ controls the size of the action space in which we are mapping, the higher value $w$ the more extreme actions will be chosen. However, in some environments, if the actions are provided only by a single expert, reducing this space can be detrimental to performance. The task specific value of $w$ is required. For ablation study over $w$ please see Appendix (A.2). Nevertheless, our approach represents a significant step toward improving the performance and stability of offline RL algorithms, and we believe it can serve as a foundation for future research in this area.

**Run time:** The code is implemented in the `PyTorch` (Paszke et al., 2019) and `JAX` frameworks and will be publicly available along with the trained networks. Our method converges within 2–3 hours on an Nvidia 1080 (12 GB) GPU. We used WanDB (Biewald, 2020) for babysitting training process. In comparison to many offline RL methods that use ensemble-based training methods, our approach provides a simpler framework with a few modifications built upon the *oracle* methods .

## 5   CONCLUSION AND DISCUSSION

In this paper, we have established a formal connection between neural optimal transport and offline reinforcement learning. Our work introduces the concept of optimal transport to deep reinforcement learning by using the action-value function and the policy as components of the OT problem between states and actions. To demonstrate the potential of our formulation, we propose an XMLR algorithm that uses the OT potential $f$ to penalize the policy and avoid inefficient actions provided in the dataset.

According to the theory, our potential $f(a)$ considers the actions from the entire expert distribution of actions. It might be reasonable to choose actions based on the state-conditional action distributions. With respect to (13), this implies that we need to modify our potential $f(a)$ by adding another input parameter to make it state-conditional: $f(s, a)$. However, our experiments show that even without state conditioning and without selecting actions from the expert's full set of actions, our method still achieves robust results. For more details please see Appendix (A.3).

Using our formulation other OT methods also can be integrated into RL. For example, various regularizations (Korotin et al., 2022b; Courty et al., 2016) and general costs (Paty & Cuturi, 2020; Asadulaev et al., 2022) can be used to incorporate task-specific information into the learned map, which can be particularly relevant in hierarchical RL problems (Bacon et al., 2017). Also, the Weak Neural OT (Korotin et al., 2022c) can be relevant in RL where stochastic behavior is preferred for exploration in the presence of multimodal goals (Haarnoja et al., 2017).

## 5.1 REPRODUCIBILITY

To reproduce our experiment we provide source code in supplementary materials. Details on used hyperparameters are presented in settings (§4.2).

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

## A APPENDIX

### A.1 PROOFS

**Proposition A.1** (Policy Improvement with Extremal Transport). *For any policy $\pi$, let's define its performance as $J(\pi) \overset{def}{=} \underset{\tau \sim \pi}{\mathrm{E}}[\sum_{t=0}^{\infty} \gamma^t r(s_t, a_t, s_{t+1})]$ where $\tau \sim \pi$ indicates the random trajectories obtained by following the policy $\pi$, $(s_0 \sim S, a_t \sim \pi(s_t), s_{t+1} \sim P(\cdot \mid s_t, a_t))$. Let $\beta$ be the policy of an expert and $\pi$ is the solution to Eq. 12 with the $Q^{\beta}$ cost function. Then it holds that:*

$$J(\pi) \geq J(\beta).$$

**Proof.** According to Kakade & Langford (2002), to compare the performance of any two policies $\pi$ and $\beta$ we can use the *Performance difference lemma*:

$$J(\pi) - J(\beta) = \frac{1}{1-\gamma} \underset{s \sim d^\pi}{\mathbb{E}} \left[ A^\beta(s, \pi) \right] \tag{17}$$

To show the improvement over the behavior policy, let's consider the difference between the behavior policy $\beta$ and the new policy $\pi$ obtained after the update of $\beta$ using our Eq.12, then we have:

$$J(\pi) - J(\beta) = \frac{1}{1-\gamma} \underset{s \sim d^\pi}{\mathbb{E}} \left[ Q^\beta(s, \pi) - V^\beta(s) \right] = \tag{18}$$

$$\frac{1}{1-\gamma} \underset{s \sim d^\pi}{\mathbb{E}} [Q^\beta(s, \underset{a \subset \mathrm{Supp}(\mathcal{A})}{\max}[Q^\beta(s, a)]) - \mathbb{E}_{a \sim \beta(s)}[Q^\beta(s, a)]] \geq 0 \tag{19}$$

Where in first we used the definition of $\pi$, and then noted that the maximum over the all actions given by the expert is always greater than the average over some actions from the experts policy $\beta$. $\square$

This result is analogous to the classic policy iteration improvements proof. The difference is that we takes max over the supp($\mathcal{A}$) rather than max over the all possible actions $A$.

### A.2 ABLATION ON THE PARAMETER $w$

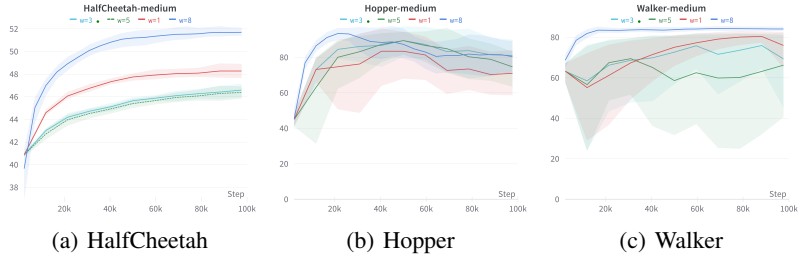

(a) HalfCheetah          (b) Hopper          (c) Walker

Figure 2: Normalized score curves of the XMRL$^{\mathrm{CQL}}$ algorithm with different values of the $w$ on MuJoCo

Table 3: Ablation study of the parameter $w$ using XMRL$^{\mathrm{CQL}}$ algorithm

|  | Dataset | $w = 1$ | $w = 3$ | $w = 5$ | $w = 8$ |
|---|---|---|---|---|---|
| | HalfCheetah | 48.2±0.6 | 47.0±0.6 | 46.5±0.3 | **51.4±0.2** |
| Medium | Hopper | 72.6±14.2 | 80.26±6.0 | 64.7±8.3 | **80.4±7.4** |
| | Walker | 65.5±25.3 | 63.9±36.8 | 73.5±11.3 | **84.3±0.6** |

The MRL formulation is hyperparameter free, applying extremal transport requires a hyperparameter $w$. But in XMRL, $w$ is a parameter that is used. This parameter influences the action selection process by determining the support range over which the policy operates. With the parameter $w$ equal to 1, our method considers all actions provided in the dataset; by increasing this value, we consider the smaller subspace of possible actions, which can be useful when the dataset is provided by a number of experts and it is necessary to select the best one from the data.

More formally, according to the (Gazdieva et al., 2023, Proposition 4), the optimal potential $f^*$ vanishes on the outliers, i.e. $f^*(a) = 0$ holds for all $a \in \text{Supp}(\mathcal{A}) \setminus \text{Supp}(\pi^* \sharp \mathcal{S})$ and the coefficient $w$ is used to control the size of this support in practice. In a perfect scenario, we want to find as few actions as possible that maximize the score, so we usually favor the higher values of $w$.

But in practice, results will vary for different datasets; for some datasets, reducing the action subspace is unnecessary. Thus, we can say that our parameter allows for nuanced control depending on the task. See the results on the MuJoCo in Figure 2 and Table 3. For the Antmaze, Pen and Door environments, see Figure 3.

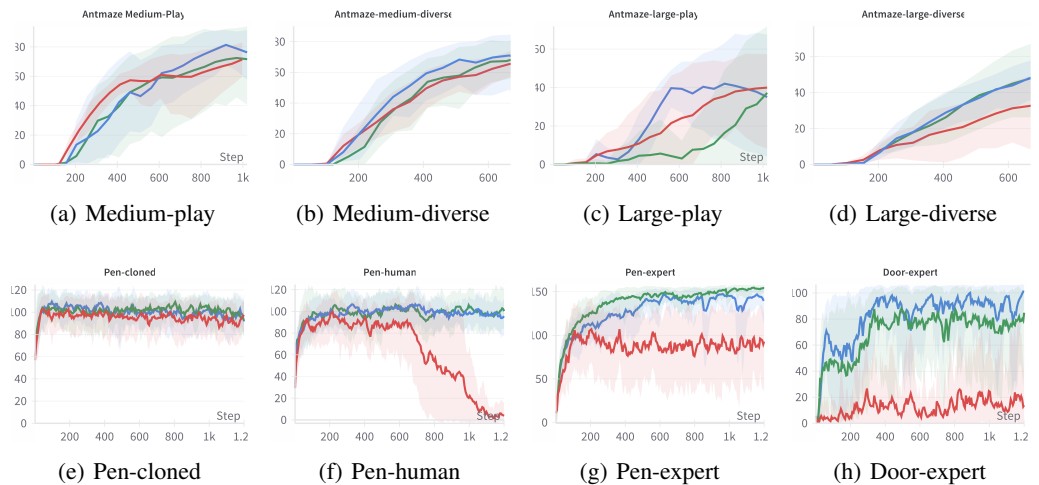

(a) Medium-play     (b) Medium-diverse     (c) Large-play     (d) Large-diverse

(e) Pen-cloned     (f) Pen-human     (g) Pen-expert     (h) Door-expert

Figure 3: Exponential moving average (coef 0.5) curves of the normalized score curves for the Antmaze, Pen, and Door tasks, XMRL$^{\text{BC}}$. Curve colors are $w = 8$, $w = 5$, $w = 1$.

.

## A.3   STATE CONDITIONAL OT FORMULATION

According to the theory, our potential $f(a)$ considers the actions from the entire expert distribution of actions. It might be reasonable to choose actions based on the state-conditional action distributions. With respect to (13), this implies that we need to modify our potential $f(a)$ by adding another input parameter to make it state-conditional: $f(s, a)$. However, our experiments show that even without state conditioning and without selecting actions from the expert's full set of actions, our method still achieves robust results. Below, we provide a side-by-side comparison of the proposed XMRL and its conditioned version.

Table 4: Comparison of XMRL$^{\text{CQL}}$ and its state-conditioned version XMRL$^{\text{CQL}}$ averaged normalized scores on `MuJoCo` tasks. Reported scores are the results of the last 10 evaluations and 5 random seeds.

|  | Dataset | XMRL$_s^{\text{CQL}}$ | XMRL$^{\text{CQL}}$ |
|---|---|---|---|
| Medium | HalfCheetah | 3.5±0.9 | 51.4±0.2 |
|  | Hopper | 83.4±9.6 | 80.4±7.4 |
|  | Walker | 63.3±32.9 | 84.3±0.6 |
| Medium-Replay | HalfCheetah | 22.0±10.1 | 44.8±0.5 |
|  | Hopper | 67.8±20.3 | 92.1±10.8 |
|  | Walker | 33.6 ±27.1 | 86.6±4.8 |
| Medium-Expert | HalfCheetah | 26.5±16.0 | 74.4±17.0 |
|  | Hopper | 59.5±24.9 | 110.2±1.9 |
|  | Walker | 49.4 ±51.0 | 111.2±0.7 |

As can be seen in Table (4), the state-conditioned version XMRL$_s^{\text{CQL}}$ does not provide an improvement in performance. We assume that this is because the state-conditioned distributions provided by the expert are not rich enough, and expanding the action space allows to act more efficiently. We can say that our marginal-based method lies between behavioral cloning and direct maximization of the

function. Recent studies have shown that direct maximization can lead to suboptimal results due to overestimation bias. Conversely, adhering too closely to the expert's policy prevents improvement. Intuitively, our extremal formulation allows us to strike a balance, maximizing by taking actions that are not radically different from those on which the function was trained.

## A.4 ADDITIONAL ILLUSTRATIONS

For completeness, we provide an additional illustration of the results of the proposed method. Specifically, in this subsection we show the normalized score curves of the CQL and our method during training.

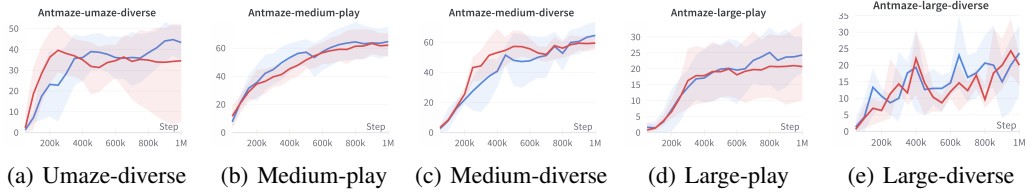

| (a) Umaze-diverse | (b) Medium-play | (c) Medium-diverse | (d) Large-play | (e) Large-diverse |

Figure 4: Normalized score curves on the Antmaze tasks, XMRL$^{CQL}$ algorithm is blue, CQL is red

On the pen and door environments Fu et al. (2020), reported in Section A.2, Figure (3) the best results of our method as follows: pen-cloned: 100.1±20, pen-human: 97.5±11, pen-expert: 154.9±4.2. door-expert: 104.9±4.5. For the environments like door-cloned and door-human, our method achieved zero scores similar to ReBRAC.

