# OpenReview forum: "Rethinking Optimal Transport in Offline Reinforcement Learning"
_ICLR.cc/2024/Conference — Submitted to ICLR 2024_

### Official Review · Reviewer_sJfc · 2023-10-21

**Soundness:** 3 good
**Presentation:** 3 good
**Contribution:** 3 good
**Rating:** 6
**Confidence:** 2

**Summary:**

This paper proposes a novel offline RL algorithm that comes from the popular optimal transport methods. Experiment results on Meta-World show that it achieves SOTA performance on various tasks.

**Strengths:**

Offline RL is recently a heated topic, and the authors propose a novel offline RL algorithm that has strong theoretical support. The idea of using optimal transport as policy regularization is novel and interesting. Experiment results are quite strong.

**Weaknesses:**

I am not quite familiar with optimal transport, so actually I did not fully check the methods. One possible weakness is that XMRL introduces an additional component $f$, which introduces additional learning complexity, especially in complex tasks.

**Questions:**

1. $f$ seems to be learned in an adversarial training manner. Is there any connection between XMRL and ATAC [1], an offline algorithm that also uses adversarial training?
2. How much additional computation complexity does XMRL introduce? The authors discuss about absolute training time, but how much more time does XMRL take compared to methods like IQL or CQL?
3. It seems that the larger $w$ is, the higher the performance. Can the authors further discuss the impact of $w$ on performance?


[1] Cheng, Ching-An, et al. "Adversarially trained actor critic for offline reinforcement learning." International Conference on Machine Learning. PMLR, 2022.

---

> ### Author Response · Authors · 2023-11-23
> **Author Response**
>
> Thank you for spending time reviewing our paper and providing valuable feedback that will help us improve the manuscript. Please find below the answers to your questions.
>
> >**Q1: $f$ seems to be learned in an adversarial training manner. Is there any connection between XMRL and ATAC [1], an offline algorithm that also uses adversarial training?.**
>
> Our method XMRL differs from ATAC by using the function $f$ for a different purpose. In XMRL, $f$ is introduced as a function to enforce constraints during the learning process, as described in Section 2, whereas in ATAC they use adversarial training to optimize the critic function $Q$ (denoted as $f$ in their paper).
>
> Moreover, although ATAC is more closely related to the CQL approach, which we compare to in our experiments, our method can use a cost/critic function trained using ATAC's method. This synergy between XMRL and ATAC could be an interesting direction for future research, showing how adversarially trained critic functions could improve constraint-based methods like ours. In the revised version of our paper, we have updated the *related works section* to include ATAC [1] as a critic penatly method.
>
> ---
>
> >**Q2: How much additional computation complexity does XMRL introduce? The authors discuss about absolute training time, but how much more time does XMRL take compared to methods like IQL or CQL?.**
>
> For the MuJoCo tasks pre-trained models were used, we trained our method for 100K steps using the pre-trained critic, the training time in the paper is reported for these settings. Comparing our method on MuJoCo Halfcheetah with training from scratch, IQL takes 10 hours, XMRL: 11 hours, CQL: 18 hours.
>
> Regarding the AntMaze tasks, we used the JAX framework, which is known for its efficiency, and the timing was as follows: for example, for the AntMaze (large-play) ReBrac takes ~60 minutes and our takes XMRL: 75.1 minutes.
>
> While introducing an additional discriminator into the training process could potentially increase the computational complexity, in reality XMRL introduces minimal overhead compared to other methods.
>
> ---
>
> >**Q3: It seems that the larger w is, the higher the performance. Can the authors further discuss the impact of w on performance??**
>
> Please, see the general response to all reviewers and revised Appendix A.2 for a detailed discussion of this topic.
>
> ---
>
> >**Concluding remarks** We truly value your reviews. Please respond to our post to let us know if the clarifications above suitably address your concerns about our work. If the responses above are sufficient, we kindly ask that you consider raising your score.

---

### Official Review · Reviewer_Z7BD · 2023-10-28

**Soundness:** 2 fair
**Presentation:** 2 fair
**Contribution:** 2 fair
**Rating:** 3
**Confidence:** 4

**Summary:**

The key idea of the paper is to compute the optimal transport between states and actions with an action-value cost function. This provides a new way to balance the two tasks in offline RL: policy improvement and distribution shift avoidance. Based on this idea, the authors propose a new algorithm called Extremal Monge Reinforcement Learning and have shown that the new algorithm outperforms BC and previous offline RLs.

**Strengths:**

1. The idea of thinking of an offline RL problem as an Optimal Transport (OT) problem is interesting. Especially considering that the Extremal Optimal Transport (ET) can be used for policy improvement.
2. The experimental results are sufficient and promising.

**Weaknesses:**

1. This contribution of the paper is insignificant, as it is barely a direct application of  Extremal Optimal Transport [1], without providing a new understanding of offline RL or solving/alleviating existing problems in offline RL.
2. The idea is not convincing. It is hard to see why we should consider an offline RL task an OT problem. Why do we want to preserve the distribution of $\mathcal{A}$ instead of considering the distribution $\mathcal{A}(s)$ or support $Supp(\mathcal{A}(s))$ independently for each state? According to my understanding, offline RL considers the distribution $\mathcal{A}(s)$ instead of the distribution of $\mathcal{A}$, as shown in W-BRAC (eq. 8).
3. XMLR does not seem better than W-BRAC. As stated in 2, the theoretical foundation of XMLR is not convincing. Furthermore, in practice, XMLR faces the same problems as W-BRAC. 1) both of them train an additional discriminator $f$. 2) both have a hyper-parameter to control the extreme of the policy, which is difficult to choose.
4. The experimental results are not surprising. According to Table 2, the results of XMRL are close to the results of ReBRAC.
5. The Writing should be improved. Typos, e.g., in the first line in section 2.4, should be fixed.

**update**
Thanks for the reply. However, I still don't think the paper is ready for publication. It is still unclear **why we should consider an offline RL task an OT problem**. Although the authors believe that XMLR can address the well-known "stitching problem", we need evidence. Besides, it is questionable whether XMLR should be considered an offline RL algorithm since it relies on Q functions estimated by other offline RL algorithms (e.g. CQL). Perhaps it should be seen as a "policy extraction" algorithm, which can be plugin into any existing offline RL. Overall, I will keep my rating.


[1] Gazdieva, Milena, et al. "Extremal Domain Translation with Neural Optimal Transport." arXiv preprint arXiv:2301.12874 (2023).

**Questions:**

1. What is the benefit of formulating an offline RL problem as an OT problem?

---

> ### Author Response · Authors · 2023-11-23
> **Author Response (Part 1)**
>
> Thank you for taking the time to review our paper and provide useful feedback. Your questions will help us improve the manuscript. Below are the answers to your questions.
>
> > **Q1: This contribution of the paper is insignificant, as it is barely a direct application of Extremal Optimal Transport [1], without providing a new understanding of offline RL or solving/alleviating existing problems in offline RL.**
>
> Our paper addresses the well-known "stitching problem" in offline RL, as detailed in **Section 3.2**. This problem has been recognized in the offline RL literature (Levine et al., 2020) as a significant obstacle to achieving strong performance in offline reinforcement learning.
>
> We believe that our contribution goes beyond a direct application of Extremal OT. We argue that without our reformulation in **Section 2**, it is unclear that Extremal OT can applicable to RL. We establish a unique connection between OT and RL domains, offering a novel perspective on the problem - **this connection has not been considered before.**
>
> Then we demonstrates the practical applicability of this novel connection by integrating recent findings in OT, specifically Extremal OT, to effectively mitigate the snitching problem. The superior performance demonstrated in our evaluation results underscores the effectiveness of our approach compared to existing methods.
>
> ---
>
> > **Q2: The idea is not convincing. It is hard to see why we should consider an offline RL task an OT problem. Why do we want to preserve the distribution of A instead of considering the distribution or support A(s) independently for each state? According to my understanding, offline RL considers the distribution A(s) instead of the distribution of A, as shown in W-BRAC (eq. 8)?**
>
> The main reason for consider an offline RL task an OT problem is because this formulation allows to solve stitching problem, and outperforms a previously proposed approaches.
>
> Regarding the choice between using $ \mathcal{A} $  and $  \mathcal{A}(s) $ , we'd like to clarify that our method is indeed flexible and can consider both $  \mathcal{A}$ and $  \mathcal{A}(s)$  distributions. In the conclusion section of our initial submission, we provided a rationale for our choice to prioritize $ A$  over $  \mathcal{A}(s) $ . This choice, was guided by our experimental observations. We found that focusing on $ f(a)$  rather than $ f(s,a)$  did not compromise performance, and we preferred this representation for its methodological simplicity.
>
> In practice we can incorporate a state-conditional information through conditional optimal transport. This simply requires adding an additional input variable to the $ f$  function to produce $f(s,a)$ .
>
> We added a separate discussion section on this topic into the revised paper. Also we have included $ f(s,a)$  results in the revised Appendix A.3. All results with $ f(s,a)$ can be reproduced with the original code provided in the Supplementary Material. Please see the networks.py file for the state-conditional version of the potential $ f(s,a)$ .
>
> ---
>
> > **Q3.1: XMLR does not seem better than W-BRAC. As stated in 2, the theoretical foundation of XMLR is not convincing. Furthermore, in practice, XMLR faces the same problems as W-BRAC. 1) both of them train an additional discriminator.**
>
> We initially did not include W-BRAC in the direct comparisons because our experiments showed that XMLR outperformed methods that in turn showed significant improvements over W-BRAC, suggesting a clear performance hierarchy.
>
> Regarding the training of an additional discriminator, we highlight an important difference in **Section 2**: W-BRAC requires *Lipschitz* constraints on the discriminator. *Lipschitz* constraints often complicate W-BRAC's discriminator training, while weight clipping, gradient penalty, and other regularizations are required. In contrast, XMLR's approach does not require Lipschitz constraints on the discriminator.
>
> ---
>
> > **Q3.2 both methods have a hyper-parameter to control the extreme of the policy, which is difficult to choose.**
>
> The role of the parameter in W-BRAC is completely different. Please see the general response to all reviewers for a detailed discussion of this topic.

---

> > ### Author Response · Authors · 2023-11-23
> > **Author Response (Part 2)**
> >
> > >**Q4: The experimental results are not surprising. According to Table 2, the results of XMRL are close to the results of ReBRAC.**
> >
> > We respectfully disagree. In the area of Offline RL, large jumps in performance have become increasingly uncommon. Achieving even modest improvements is often challenging, yet these improvements can still represent meaningful progress. This is particularly true in the environments we have examined, where many sophisticated methods, like IQL, ReBRAC, etc, have already set a high-performance benchmark.
> >
> > Furthermore, a comparison of our XMRL method with the existing SOTA OT-based method OTR provides essential context for our claims. Table 2 in our paper shows that OTR, which uses the IQL backbone, improves on IQL in 2/6 environments, while XMRL consistently improves on CQL in every environment and improves on ReBRAC in 4/6.
> >
> > ---
> >
> > >**Q5:The Writing should be improved. Typos, e.g., in the first line in section 2.4, should be fixed..**
> >
> > Thanks for pointing this out, typos are fixed in the revised version.
> >
> > ---
> >
> > >**Q6: What is the benefit of formulating an offline RL problem as an OT problem?**
> >
> > Please see the answers to the Q1-4.
> >
> > ---
> >
> > >**References:**
> > >Sergey Levine, Aviral Kumar, George Tucker, and Justin Fu. Offline reinforcement learning: Tutorial, review, and perspectives on open problems, 2020.
> >
> > >**Concluding remarks:** In conclusion, we truly value your reviews. We hope that the revisions and clarifications will influence and improve your overall opinion. If we've managed to resolve your principal concerns and questions, we'd be thankful for your endorsement through an elevated score of our submission. On the other hand, if there are remaining issues or questions on your mind, we're more than willing to address them.

---

### Official Review · Reviewer_wDEF · 2023-11-01

**Soundness:** 2 fair
**Presentation:** 3 good
**Contribution:** 2 fair
**Rating:** 6
**Confidence:** 3

**Summary:**

In offline reinforcement learning, optimal transport can be used to assess the distance between a certain policy and an expert policy. When using the Wasserstein distance, this leads to the W-BRAC algorithm from [Wu et al, 2022]. This paper overcomes some drawbacks of BRAC and related methods (choice of regularizing hyperparameter and Lipschitz constraints) by looking at offline RL as a saddle-point problem that arises from the dual form of the Kantorovich problem, and leveraging standard duality machinery. To avoid inefficient actions (stichting between “good” trajectories in the dataset), set inclusion constraints are replaced by inequality constraints, providing extremal transport in some cases. The proposed method, which appears to be sensitive to a regularizing hyperparameter, provides improvements over relevant baselines in the MuJoCo and Antmaze benchmarks.

**Strengths:**

- Novel formulation of offline RL using standard duality results and OT machinery.
- The empirical evaluation contains 9 Mujoco and 6 Antmaze environments against relevant benchmarks. Although performance improvements are not significant in some environments, the proposed method is almost consistently among the top 3. By comparing XMRL with BC and CQL alone, the impact of the proposed approach is disentangled from that of the techniques used to avoid overestimation bias.
- The paper is clearly written and, to the best of my knowledge, the related work is adequately described.

**Weaknesses:**

- I do not see the relevance of proposition 3.1 (policy improvement), since it does not appear to give any insight about the method (the proposition is not cited/used anywhere in the paper) and the proof is basically the same as the one for classic policy iteration improvement.
- Evaluating the performance of the method in more challenging environments or tasks would enrich the experimental section. The impact of entropy regularization on the performance of the method is not assessed.
- The method seems considerably sensitive to the parameter w (as seen in appendix 2), so one of the drawbacks of W-BRAC is still present in this method.
- Typos: “Fol all experiments” section 4.2.

**Questions:**

-

---

> ### Author Response · Authors · 2023-11-23
> **Author Response**
>
> Thank you for your detailed analysis of our paper and your thoughtful suggestions for improvement. All typos have been corrected. Below is our response to your comments.
>
> > **Q1:I do not see the relevance of proposition 3.1 (policy improvement), since it does not appear to give any insight about the method (the proposition is not cited/used anywhere in the paper) and the proof is basically the same as the one for classic policy iteration improvement.**
>
> Thank you for raising your concern that **Proposition 3.1** is not cited in the paper. The primary purpose of **Proposition 3.1** is to emphasize the focus of our method on policy improvement rather than behavior cloning, as stated in the Introduction and **Section 3**. We mentioned our proposition now in **Sections 1, and 3**.
>
> ---
>
> > **Q2:Evaluating the performance of the method in more challenging environments or tasks would enrich the experimental section.**
>
> In response to your suggestion, we conducted experiments on the Pen and Door environments of the D4RL. Our XMRL-BC method showed performance similar to the ReBRAC method, specifically: **pen-expert: 154.9±4.2, pen-cloned: 100.1±20, pen-human: 97.5±11, door-expert: 104.9±4.5**. For the environments like door-cloned and door-human, our method achieved zero scores similar to ReBRAC. We have included these results in *Appendix A.4* and additionally provided an ablation study on parameter $w$ for these environments in **A.2**.
>
> ---
>
> > **Q3:The impact of entropy regularization on the performance of the method is not assessed?**
>
> We did not perform an ablation study on entropy regularization as *it is not part of our contribution*. Entropy regularization was only applied in the context of CQL-based experiments performed on the Antmaze environments, while the CQL-backbone includes entropy regularization by design. Conversely, for the *simpler* MuJoCo environments, we opted for a version of CQL Eq.(14) without any regularization.
>
> ---
>
> > **Q4: The method seems considerably sensitive to the parameter w (as seen in appendix 2), so one of the drawbacks of W-BRAC is still present in this method.**
>
> This is not true. The drawbacks of W-BRAC are not present in our method, while the role of our parameter is completely different. The fact that a hyperparameter (even a different) is required is noted as a *limitation* in the **(Section 4.3)** of the initial submission. Please see the general response to all reviewers for a detailed discussion of this topic.
>
> ---
>
> >**Concluding remarks**: Please respond to our post to let us know if the clarifications above suitably address your concerns about our work. If you finds the responses above are sufficient, we kindly ask that you consider raising your score.

---

### Author Response · Authors · 2023-11-23
**General answer to all reviewers**

Dear reviewers, thank you for your valuable feedback and suggestions to improve the paper. We appreciate that reviewers find our idea as interesting (**Z7BD, sJfc**), novel (**sJfc**) with clearly presented (**wDEF**). We are also glad to see that reviewers recognize a consistent (**wDEF)**, sufficient, (**Z7BD**) and SOTA scores achieved by our method during practical evaluation (**sJfc**). Please find the answers to your questions below.

> ###  Parameter $w$

First, it is important to note that in our original submission (**Section 4.3**), we acknowledged that the dependence on $w$ does exist, and we identified it as a *limitation*. However, we want to note that the MRL algorithm does not use hyperparameters.

The discussions on $a$ in Section 2.4 and then in Section 3.1 are given to show that the previously proposed methods include OT in the RL as an additional loss with weight $a$ and our formulation does not. Our formulation is to integrate the optimal transport directly, without any weight, considering the entire RL problem as OT.


**This represents a fundamental conceptual shift**, rather than a mere difference in parametrization, **thus the roles of the parameters are completely different**, and the drawbacks of considering OT as an additional loss associated with hyperparameters in W-BRAC are avoided.

In XMRL, $w$ is a parameter that influences the action selection process by determining the support range over which the policy operates. With the parameter $w$ equal to 1, our method considers all the actions provided in the dataset; by increasing this value, we consider the smaller subspace of possible actions, which can be useful when the dataset is provided by a series of experts and it is necessary to select the best one from the data.

This crucial distinction leads us to consider $w$ as more than just a hyperparameter - it is a core element of our model that allows fine-tuning of the policy improvement. While the provided offline datasets are different, the results may vary for different datasets; for some datasets, reducing the action subspace is unnecessary.

To further address concerns about the impact of the $w$ parameter, we have expanded our paper with additional experiments and discussion. This new ablation study, detailed in the extended appendix, systematically examines the influence of $w$, allowing for a deeper understanding of its role in performance variance across environments. Please see **Appendix A.2** for more details.

 > ### Revised version

The revised revision of our paper has been uploaded, addressing the comments and questions raised by the reviewers. In summary, the revised paper contains the following updates, all of which are highlighted in blue for reviewers' convenience:
 - Included more results on the $w$ parameter ablation study (**Appendix A.2) (wDEF, Z7BD, sJfc)**
- Added a subsection discussing results with state-conditional formulation (**Appendix A.3**) **(Z7BD)**
- Provided a necessary citation, clarified statements, and fixed a typo.

We hope that these revisions address the reviewers' questions and improve the clarity of the paper.

---

### Meta-Review · Area_Chair_Z4Vg · 2023-12-09

**Metareview:**

The authors propose an approach to offline RL based on optimal transport (OT). It is based on the idea that the optimal transport between states and actions, which can serve as a policy, can be computed using an action-value cost function. Using this idea, they formulate the offline RL as an extremal optimal transport problem and derive an algorithm, called Extremal Monge RL (XMRL), to solve it. The authors claim that their algorithm not only addresses the distribution shift problem, but can go beyond the behavior policy. They evaluate their algorithm on continuous control problems and compare it with existing methods.

Establishing a connection between OT and RL can potentially be helpful in deriving new RL algorithms and addressing issues such as stitching problem in offline RL. The formulation of offline RL as an extremal OT problem and the experimental results are promising.

However, the paper is hard to follow and its goal and contributions are not clear. Some of the derivations require better explanation and clarification. More importantly, the authors do not make it clear which of the offline RL problems/difficulties do they expect to address using OT formulation, and why/how does this new formulation address it? What are the offline RL problems that are being addressed? Why/how does this new optimization problem (algorithm) address them and why have they not been addressed by the existing methods? The last question needs to be answered using both the optimization problem and proper empirical evaluation (empirical evidence)? Another aspect of the paper that needs improvement is proper comparison (complexity, performance, etc.) with the existing algorithms? Is it a simpler algorithm to work with (more/less parameters to tune, more/less components to learn, etc.)? The authors added more results on parameter w, but there is no conclusion or recipe on when (for which problems) is necessary to reduce the action subspace (using w). No insight on how this parameter should be tuned. There are good results in the paper, both algorithmically and empirically, but they have not been properly fleshed out to have a thorough evaluation of the entire work.

**Justification For Why Not Higher Score:**

The paper is hard to follow and its goal and contributions are not clear. Some of the derivations require better explanation and clarification. More importantly, the authors do not make it clear which of the offline RL problems/difficulties do they expect to address using OT formulation, and why/how does this new formulation address it? What are the offline RL problems that are being addressed? Why/how does this new optimization problem (algorithm) address them and why have they not been addressed by the existing methods? The last question needs to be answered using both the optimization problem and proper empirical evaluation (empirical evidence)? Another aspect of the paper that needs improvement is proper comparison (complexity, performance, etc.) with the existing algorithms? Is it a simpler algorithm to work with (more/less parameters to tune, more/less components to learn, etc.)? The authors added more results on parameter w, but there is no conclusion or recipe on when (for which problems) is necessary to reduce the action subspace (using w). No insight on how this parameter should be tuned. There are good results in the paper, both algorithmically and empirically, but they have not been properly fleshed out to have a thorough evaluation of the entire work.

**Justification For Why Not Lower Score:**

None

---

### Decision · Program_Chairs · 2024-01-16

Reject